# TLRs Gene Polymorphisms Associated with Pneumonia before and during COVID-19 Pandemic

**DOI:** 10.3390/diagnostics13010121

**Published:** 2022-12-30

**Authors:** Svetlana Salamaikina, Maria Karnaushkina, Vitaly Korchagin, Maria Litvinova, Konstantin Mironov, Vasily Akimkin

**Affiliations:** 1Central Research Institute of Epidemiology, Federal Service for Surveillance on Consumer Rights Protection and Human Wellbeing Russian Federation, Novogireevskaya 3a Str., 111123 Moscow, Russia; 2Medical Faculty, The Peoples’ Friendship University of Russia (RUDN University), Miklukho-Maklaya Str. 6, 117198 Moscow, Russia; 3Department of Medical Genetics, I.M. Sechenov First Moscow State Medical University (Sechenov University), 8-2 Trubetskaya Str., 119991 Moscow, Russia

**Keywords:** pneumonia, COVID-19, TLR, SNP, real-time PCR

## Abstract

Background: The progression of infectious diseases depends on the characteristics of a patient’s innate immunity, and the efficiency of an immune system depends on the patient’s genetic factors, including SNPs in the *TLR* genes. In this pilot study, we determined the frequency of alleles in these SNPs in a subset of patients with pneumonia. Methods: This study assessed six SNPs from *TLR* genes: rs5743551 (*TLR1*), rs5743708, rs3804100 (*TLR2*), rs4986790 (*TLR4*), rs5743810 (*TLR6*), and rs3764880 (*TLR8*). Three groups of patients participated in this study: patients with pneumonia in 2019 (76 samples), patients with pneumonia caused by SARS-CoV-2 in 2021 (85 samples), and the control group (99 samples). Results: The allele and genotype frequencies obtained for each group were examined using four genetic models. Significant results were obtained when comparing the samples obtained from individuals with pneumonia before the spread of SARS-CoV-2 and from the controls for rs5743551 (*TLR1*) and rs3764880 (*TLR8*). Additionally, the comparison of COVID-19-related pneumonia cases and the control group revealed a significant result for rs3804100-G (TLR2). Conclusions: Determining SNP allele frequencies and searching for their associations with the course of pneumonia are important for personalized patient management. However, our results need to be comprehensively assessed in consideration of other clinical parameters.

## 1. Introduction

Pneumonia is an acute infectious disease characterized by focal lesions in the respiratory portions of the lungs with interalveolar exudation. According to the World Health Organization (WHO), lower respiratory tract infections, including pneumonia, are the fourth leading cause of death worldwide [1,2].

Pneumonia is associated with individual characteristics, including age and chronic conditions. The leading risk factors for severe pneumonia are smoking, obesity, alcohol abuse, and air pollution [3].

The most common agents of pneumonia are Gram-positive and Gram-negative bacteria, fungi and respiratory viruses. The COVID-19 pandemic had a significant impact on the etiological structure of community-acquired pneumonia. Against the spread of SARS-CoV-2, an increased incidence of *S. pneumoniae-* and *H. Influenzae*-associated pneumonia was observed [4]. The Russian Federation updated their clinical guidelines for the diagnosis of community-acquired pneumonia in adults in 2021 [5]. The protocol takes into account international experience and WHO recommendations.

Bacterial colonization of the lower respiratory tract affects host immunity and is one of the factors responsible for predisposition to pneumonia. When low-virulence microbes penetrate the lungs, resident innate immune defense mechanisms, such as Toll-like receptors (TLRs) provide effective defense [6,7]. TLR activation is important for the initiation and regulation of T cell innate immune response.

The first gene of this family to be studied was *TLR4*. The genetic variation of *TLR4* is associated with the receptor for bacterial lipopolysaccharide (LPS) and a central intracellular signal transducer (TIRAP/Mal). TIRAP/Mal affects cytokine release and is responsible for susceptibility to severe hospital-acquired infections and its course in distinct patient populations [8]. TLR4 was also reported to recognize the spike (S) protein of SARS-CoV-2 and to activate NF-κB signaling to produce IL-1β [9].

Infectious diseases caused by the genetic features of human immune system response could be referred to as genetic diseases. According to genome-wide association studies (GWAS), some single nucleotide polymorphisms (SNP) of the TLR gene family could be associated with a predisposition to pneumonia.

The study by Hongxiang Lu et al. showed a significant association of rs5743551 (*TLR1*) with the development of sepsis in patients admitted to the intensive care unit [10,11]. The association between genomic variants and a risk of respiratory infections or the severity of these conditions with a special focus on SARS-CoV-2 has shown that rs5743551 is associated with higher TLR1-induced NF-κB activation; higher cell surface *TLR1* expression; and increased susceptibility to organ dysfunction, death, and Gram-positive infection in sepsis [12].

Polymorphisms in the *TLR2* and *TLR4* genes are mostly associated with a susceptibility to tuberculosis [13,14]. In addition, a pilot study concluded that the rs5743708 (*TLR2*) genotype may be a risk factor for sepsis in adult patients [15]. A meta-analysis studied by Jun-wei Gao et al. provided evidence of a direct effect of the *TLR2* Arg753Gln (rs5743708) polymorphism on sepsis risk, especially in people of European descent [16]. rs5743708 may be the sole contributor to the association of the *TLR2* gene with pneumonia [17]. In another study, genotype CC of rs3804100 (*TLR2*) was associated with a significantly increased risk of developing severe sepsis [18].

Genetic variability in the *TLR4*, *CD14*, and *FcγRIIA* genes is associated with an increased risk of developing an invasive disease in patients with a severe case of pneumonia [19,20,21,22]. The rs5743810 polymorphism in the *TLR6* gene is closely related to the rs5743551 polymorphism in the *TLR1* gene, which has been shown to be closely related to the severity of pneumonia and sepsis [23]. The association of this polymorphism with tuberculosis was shown in a meta-analysis by C. Skevaki et al., similar to polymorphisms in the *TLR2* and *TLR4* genes [24].

The effect of rs3764880 *TLR8* was also shown to be closely related to Gram-positive bacteria [25]. Alessia Azzarà et al. showed an association of this polymorphism with COVID-19 outcomes in Italian families [26].

The identification of factors affecting severe or complicated pneumonia courses (including genetic factors) is an urgent task in the current field of medicine. Real-time PCR provides a fast and reliable determination of SNPs in samples of biological material obtained during routine laboratory tests in the clinic.

The aim of our study is to compare the allele frequencies of genetic polymorphisms in *TLR* genes in samples obtained from individuals with pneumonia in the period before the appearance of SARS-CoV-2 and during the COVID-19 pandemic.

## 2. Materials and Methods

### 2.1. Study Design

Depersonalized whole blood samples from pneumonia patients were used in this study. The samples were taken in 2019 and in 2021 from patients admitted to the intensive care unit of M.F. Vladimirsky Moscow Regional Clinical Hospital. Patients with a confirmed diagnosis of pneumonia were enrolled into the Case 1 group in 2019. The pneumonia was diagnosed based on the confirmed presence of a bacterial pathogen corresponding to the clinical guidelines for out-of-hospital pneumonia in adults. Bacterial isolations from tracheobronchial aspirate were performed. A sterile vacuum aspirator connected to an endotracheal tube was used to obtain the tracheal aspirate. No less than 1 mL of the aspirate was obtained from the tracheobronchial tree and collected in a sterile container. The presence of bacterial contagious agents was confirmed for all patients participating in this study.

Patients with a confirmed diagnosis of COVID-19 were enrolled in the Case 2 group in 2021. A diagnosis of COVID-19 was confirmed by a positive PCR test for SARS-CoV-2, according to the current WHO temporary guidelines for 2021. Case 2 samples were collected from patients with bacterial pneumonia developed on the background of COVID-19.

This study also included healthy people without any infectious disease symptoms as a control group (enrolled in 2021).

All subjects gave their informed consent for inclusion before they participated in this study. This study was conducted in accordance with the Declaration of Helsinki, and the protocol was approved by the Ethics Committee of M.F. Vladimirsky Moscow Regional Clinical Hospital (Project identification code No.5 14.05.2019).

The inclusion criteria were written informed consent to participate in the study, age from 18 to 70 years, and an established diagnosis of community-acquired pneumonia.

The exclusion criteria were confirmed diagnosis of HIV, hepatitis B or C, bronchial asthma, cystic fibrosis, tuberculosis, bronchiectasis, interstitial lung diseases, lung cysts, occupational lung diseases, oncological diseases, alpha1-antitrypsin deficiency, or pneumothorax. An additional exclusion criteria for the Case 2 group was an increased level of procalcitonin [27].

Standard clinical–laboratory examinations and computed tomography (CT) scans of the thoracic organs, microbiological examinations of tracheobronchial aspirate, and blood test (biochemical blood tests for determining the C-reactive protein, procalcitonin, and D-dimer levels) were carried out in all patients [28]. We evaluated the severity of pneumonia using the validated PORT and SOFA scales [29,30].

The study design is presented in Figure 1.

### 2.2. SNP Selection

The selection of SNPs in this study was based on recently data. The selected SNP in the *TLR* genes associated with sepsis and respiratory tract diseases are in Table 1.

### 2.3. Oligonucleotides Design

Real-time PCR primers and probes were designed for appropriate alleles using reference sequences from the SNP database [31]. We followed the design guidelines recommended by the instrument manufacturer (Qiagen, Hilden, Germany) [32]. The annealing temperature for the probes was adjusted to 60 °C. The primer and probe concentrations were chosen empirically and ranged from 0.2 to 0.4 and from 0.06 to 0.12 µM, respectively. The oligonucleotide sequences are present in Table 2. The letters in the probe names reflects the SNP allele.

### 2.4. DNA Extraction and Real-Time PCR Conditions

Genetic studies (DNA isolation and PCR) were carried out using blood samples collected during standard clinical and laboratory studies (general or biochemical blood analysis). All used reagents and kits were made in the Central Research Institute of Epidemiology, Federal Service for Surveillance on Consumer Rights Protection and Human Wellbeing Russian Federation (AmpliSens, Moscow, Russia).

DNA extraction was performed with the “RIBO-prep” kit (AmpliSens, Russia) in whole blood samples.

The PCR tests were carried out in a final volume of 25 μL, containing 10 μL of extracted DNA; 10 μL of dNTP (0.44 mM), the primers, and the probes; 4.5 μL of RT-PCR-mix-2 FEP/FRT reagents; and 0.5 μL of TaqF polymerase. The real-time PCR amplification was performed using the Rotor Gene Q cycler (Qiagen, Germany) with the following conditions: 95 °C for 15 min (1 cycle); 95 °C for 5 s/60 °C for 20 s/72 °C for 10 s (45 cycles with fluorescent signal detection at 60 °C). In the amplification results analysis, the threshold was set at 10% of the highest fluorescent signal value for each channel.

The real-time PCR results confirmation were performed by PyroMark Q24 sequencing using the reagent kits recommended by the equipment manufacturers (Qiagen, Hilden, Germany).

### 2.5. Statistical Analysis

The genotyping results from the studied groups were compared with the EUR population characteristics derived from the 1000 genome project, which includes genotype data from five caucasoid populations: CEU (Utah population (CEPH) with Northern and Western European ancestry), FIN (Finnish populations in Finland), GBR (British populations in England and Scotland), IBS (Iberian populations in Spain), and TSI (Toscani population in Italia) [33]. A statistical analysis was performed using R (version 4.2.1), including functions for the analysis of contingency tables: Pearson’s χ^2^ test and Fisher’s exact test. The Holm–Bonferroni approach was used to adjust for multiple comparisons. Association and risk scores (ORs) for a specific genetic risk model (codominant (major allele homozygotes vs. minor allele homozygotes) and (major allele homozygotes vs. heterozygotes), dominant (major allele homozygotes vs. heterozygotes+minor allele homozygotes), recessive (minor allele homozygotes vs. major allele homozygotes+heterozygotes), overdominant (heterozygotes vs. major allele homozygotes+minor allele homozygotes), and log-additive (major allele homozygotes vs. heterozygotes vs. minor allele homozygotes)) were performed using the SNPassoc [34], and epitools [35] packages. The results were considered statistically significant at *p* < 0.05. The genetic characteristics of the samples were analyzed using the adegenet [36] and HardyWeinberg [37] packages. All graphical results were generated using the ggplot2 [38] and ggstatsplot [39] packages.

## 3. Results

### 3.1. Epidemiological Characteristics of Studied Groups

This study included 161 patients with pneumonia and 99 healthy individuals. The size of the Case 1 group was 76. The mean age of the patients was 43.89 ± 14.54 years. We had 85 patients in the Case 2 group, with an average patient age of 65.87 ± 13.63 years.

The control group consisted of 99 people, and the average age was about 39.76 ± 12.12 years. The significant differences in age for patients with viral and bacterial pneumonia are presented in Table 3.

The groups are characterized by a non-equal gender representation. Males dominated the Case 1 group, whereas females dominated the Case 2 and control groups. Differences in the sex and age distributions of the samples in the study groups are shown in Figure 2.

The sex and age distributions in the groups were not uniform. Age in the Case 1 group was lower than that in the other studied groups. Additionally, in the Case 2 group, an overwhelming majority of patients were women. Stratification by sex was not possible due to the consequent sample reduction and lack of statistical support for the analysis.

### 3.2. Genotyping Results

The analysis of allele frequencies in the Case 1, Case 2, and control groups shows no significant differences from the EUR population. Table 4 summarizes the significant differences in the frequency of minor alleles in the studied groups and in the EUR population. Only one SNP differed significantly from the Hardy–Weinberg equilibrium in the bacterial pneumonia sample, rs3804100 (*p* = 0.04), but did not differ significantly in the control sample (*p* = 1.00). The results of the comparison of allele and genotype frequencies against the EUR population for all studied SNPs are presented in Appendix A, respectively.

A significant result was obtained for rs3764880 when comparing the Case 1 and the control groups. Since rs3764880 (*TLR8*) is located on the X chromosome, females have the potential to be heterozygous, whereas in males, the locus can be represented only by one allele, A or G. In the analysis of groups containing both females and males samples, we used a recessive model to estimate risk. The model assumes that the at-risk allele manifests only in the homozygous or hemizygous state.

The group of at-risk allele carriers included females with the GG genotype and males with a single G allele. Females with the AA and AG genotypes and males with the A allele made up the comparison group. The Case 1 group had a higher number of the minor allele carriers than the control group. A significant difference in the genotypic distribution between patients and controls was found for GG vs. AA + AG, *p* = 0.00012 (Table 5).

The rs5743551-G allele probably has a protective effect against the risk of bacterial pneumonia (OR = 0.18). The observed associations, though, do not reach statistical significance (*p* > 0.05) due to the low frequency of rare alleles and the insufficient group sizes.

Significant differences between the control and Case 2 groups were observed for rs3804100 (*TLR2*) using the log-additive model (OR = 0.32(0.11–0.93); *p* = 0.025). The estimates for the models of each SNP are presented in Appendix A.

The association was observed only for two polymorphisms in TLR genes with the occurrence of pneumonia not associated with COVID-19 (Case 1) in the studied samples. Individuals harboring the rs5743551-A and rs3764880-G alleles may be more susceptible to the occurrence of pneumonia compared with carriers of the alternative allele.

## 4. Discussion

Innate immunity has a significant role in the immunopathogenesis of inflammatory responses, especially during response to pathogens of various kinds of community-acquired pneumonia and associated sepsis [40]. TLRs are expressed in immune cells, fibroblasts, and epithelial cells, including type II pneumocytes, which highly express *ACE2* in the airways [41,42,43]. The examined data show a frequent association with adverse outcomes in both viral and bacterial lower respiratory tract diseases. Studying the SNP allele associations in genes involved both innate and acquired immunity shows that this is likely to play a significant role in the entire course of the disease. The genetic predisposition factors of the host organism, as well as pathogen characteristics, have to be considered. Relying on the analyzed data, the SNPs listed in Table 1 were selected to compare the genetic features of the groups with bacterial and viral pneumonia.

In accordance with our previous results, the rs5743551 (*TLR1*) *GG* and *GA* genotypes are prognostic markers associated with the risk of severe community-acquired pneumonia [28]. However, the group sizes should be increased for reliable and reproducible results. Wurfel et al. showed that the rs5743551-G allele variant is associated with increased cell surface expression of *TLR1* and with subsequently elevated TLR1 mediated cytokine production leading to higher TLR1-induced NFκB activation [44]. This raises questions about the effect of increased TLR1 expression in enhancing severe pulmonary diseases.

TLR polymorphism with a significant association with pulmonary tuberculosis was present in *TLR1* rs5743572 in intron, *TLR2* rs3804100 in exon, and *TLR6* rs5743808 in exon and among MDR-TB isolates from patients with pulmonary MDR-TB of a severe and moderate/mild degree [45]. Genetic variations in *TLR2*, *TLR4*, and *TLR6* affect the binding of the bacterial LPS receptor and the central intracellular signal transducer (TIRAP/Mal). The SNPs in the studied TLR genes demonstrated influences on cytokine release, susceptibility rates and the course of severe hospital-acquired infections in different patient groups [46]. Our results confirm that rs3804100 (*TLR2*) has an essential role in the host’s immune defense. Previous studies found significant associations between higher measles-neutralizing antibodies and susceptibility to tuberculosis infection [47,48].

Studies have shown that rs4986790-G (*TLR4*) and rs3764880-G (*TLR8*) convey altering the balance of the NF-κB and type I IFN axes and that this interaction plays a crucial role in a successful host response against Mtb [49]. Signaling through TLR8 involves signaling via the IRAK pathway and activation of the inflammatory signal within the cell. TLR8 is found in the lung, could lead to a cytokine storm in SARS-CoV-2, and could cause different side effects [50]. Siv H. Moen et al. showed the effect of a TLR8 inhibitor on IL-12p70 and IL-1β production against *P. aeruginosa* infection. The inhibition effect was shown in a rather variable range depending on the strain and conditions [51].

It has been shown that SARS-CoV-2 RNA is capable of stimulating TLR8-dependent inflammatory response in macrophages in vitro [52]. Some other researchers have suggested that TLR8 has less clinical relevance in COVID-19 infections [53]. Similarly, our results show the significance of the GG rs3764880 (*TLR8*) genotype in Case 1 group, rather than in Case 2 group. Our findings support the fact that the investigated polymorphism is not as clinically important in pneumonia with a confirmed diagnosis of COVID-19.

The present results demonstrated significant differences between groups of patients with pneumonia before and after the emergence of SARS-CoV-2. The methods we developed are based on the use of real-time PCR for SNP allele detection. An important feature of this study is our finding that rs5743551-A and rs3764880-G might be risk factors for the Case 1 group compared to the control group. At the same time, rs3804100-T might be a risk factor for the Case 2 group compared to the control group. The exact underlying molecular mechanisms of our findings remain to be explored by further experimental studies.

## 5. Conclusions

The study of genetic risks in the population is important for individualized diagnosis and prediction of complicated course of infections. This research makes it possible to assess population risks of pneumonias from an epidemiological point of view. It will allow us to define additional risk monitoring parameters and increase the efficiency of infectious disease surveillance. Studies of genetic susceptibility to pneumonias are an important issue, which can be applied together with diagnostics to improve the efficiency of personalized approaches to the management of pneumonia patients.

According to our findings, individuals harboring the rs5743551-A and rs3764880-G alleles may be possible risk factors for vulnerability to bacterial pneumonia.

A limitation of this study is that it did not analyze several important parameters of the course of infectious diseases. We did not consider the pathogens’ virulent properties: the etiology of bacterial pneumonia and SARS-CoV-2 gene variants. Further research considering these properties and other characteristics, including epidemiological ones, should be carried out. The results of the study need to be confirmed on a larger independent sample and the list of studied polymorphisms in the TLR genes needs to be expanded to obtain reliable results.

## Figures and Tables

**Figure 1 diagnostics-13-00121-f001:**
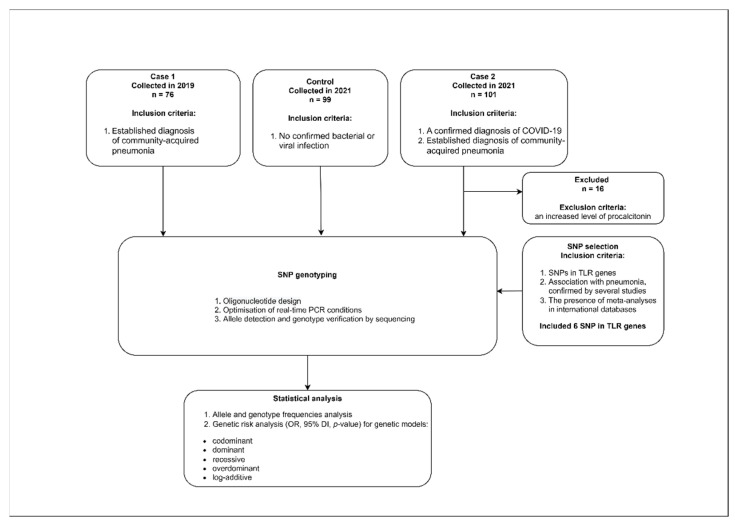
Study design.

**Figure 2 diagnostics-13-00121-f002:**
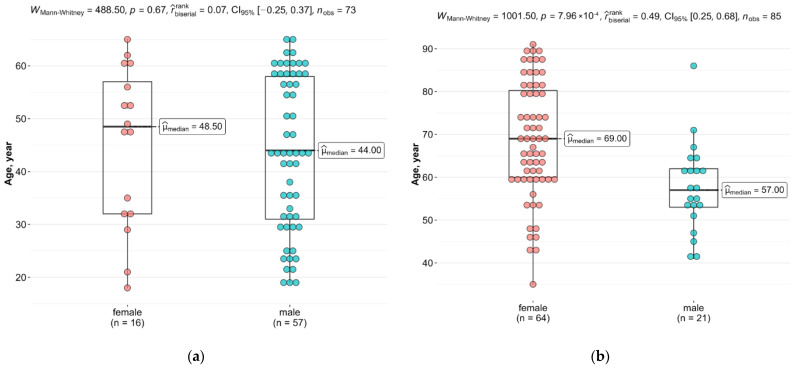
The distribution of samples by gender and age: (**a**) Case 1 group; (**b**) Case 2 group.

**Table 1 diagnostics-13-00121-t001:** Selected SNPs in TLR genes associated with pathological conditions.

Gene, SNP	Risk Allele	Associated Condition
*TLR1*, rs5743551	G	Sepsis
*TLR2*, rs5743708	A	Sepsis, tuberculosis
*TLR2*, rs3804100	T	Tuberculosis, measles-specific antibody levels following immunization
*TLR4*, rs4986790	G	Sepsis, tuberculosis
*TLR6*, rs5743810	G	Tuberculosis, allogeneic hematopoietic stem cell transplant recipients
*TLR8*, rs3764880	G	SARS-CoV-2 clinical outcomes, tuberculosis

**Table 2 diagnostics-13-00121-t002:** Oligonucleotide characteristics.

Gene, SNP	Name	5′–3′ Sequence
*TLR1*, rs5743551	TLR1-F	AAT CCA GGC TGA GGG AGC AA
	TLR1-R	ACT GGG AGT AGT GGG GAT GAG T
	TLR1-A	(FAM)GCTGAGAAGCTTCC(BHQ1) ^1^
	TLR1-G	(R6G)GCTGAGGAGCTTCC(BHQ1) ^1^
*TLR2*, rs5743708	TLR2-F	CATTCTTCTGGAGCCCATTGAGA
	TLR2-R	CGCAGCTCTCAGATTTACCCAA
	TLR2-G	(FAM)GCTGCGGAAGAT(BHQ1) ^1^
	TLR2-A	(R6G)AAGCTGCAGAAGAT(BHQ1) ^1^
*TLR2*, rs3804100	TLR2-F1	CTGAAACTTGTCAGTGGCCAGAA
	TLR2-R1	TTCCAGTGTCTTGGGAATGCA
	TLR2-C	(FAM)TACACAGCGTAACAG(BHQ1) ^1^
	TLR2-T	(R6G)TACACAGTGTAACAG(BHQ-1) ^1^
*TLR4*, rs4986790	TLR4-F	GGCCTGTGCAATTTGACCATT
	TLR4-R	GTCACACTCACCAGGGAAAATGA
	TLR4-A	(FAM)CCTCGATGATATTATTG(BHQ1) ^1^
	TLR4-G	(R6G)CCTCGATGGTATTATTG(BHQ1) ^1^
*TLR6*, rs5743810	TLR6-F	CTATGTGGTTGAGGGTAAAATTCAGTAAG
	TLR6-R	TGAATGATGACAACTGTCAAGTTTTC
	TLR6-A	(FAM)AAGGTTGAACCTCTG(BHQ1) ^1^
	TLR6-G	(R6G)AGGTTGGACCTCTG(BHQ1) ^1^
*TLR8*, rs3764880	TLR8-F	GCTGCTGCAAGTTACGGAATGAA
	TLR8-R	GGGTCAGAAACCCCATATTCTGTT
	TLR8-A	(FAM)CAGAAACATGGTAAG(BHQ1) ^1^
	TLR8-G	(R6G)CAGAAACGTGGTAAG(BHQ1) ^1^

^1^ The probes contain from 4 to 7 locked nucleic acids.

**Table 3 diagnostics-13-00121-t003:** Demographic characteristics of the study subjects.

Variables	Patients (n = 161)			Controls (n = 99)	*p*
	Case 1 (n = 76)	Case 2 (n = 85)	*p*		
Age, mean (SD), y	43.89 (14.54)	65.87 (13.63)	<0.0001	39.76 (12.12)	<0.0001
Male, n (%) *	57 (75)	21 (24.7)	<0.0001	24 (24.2)	0.0001

* Sex differences between groups were tested by Pearson’s χ^2^ test. SD: standard deviation.

**Table 4 diagnostics-13-00121-t004:** Significant differences between the studied groups and EUR population.

Samples	SNP	Minor Allele Frequency (MAF)	HWE *, *p*	Fisher’s Exact Test for Equality of Allele Frequencies, *p*
Case 1	rs5743551 (*TLR1*)	G (0.243)	0.056	0.332
	rs3804100 (*TLR2*)	C (0.072)	**0.040**	0.723
	rs4986790 (*TLR4*)	G (0.079)	0.058	0.271
	rs3764880 (*TLR8*)	G (0.274)	0.360 **	0.903
Case 2	rs5743551 (*TLR1*)	G (0.194)	0.074	**0.015**
	rs3804100 (*TLR2*)	C (0.029)	1.000	0.110
	rs4986790 (*TLR4*)	G (0.100)	0.588	**0.040**
	rs3764880 (*TLR8*)	G (0.235)	0.519 **	0.417
Control	rs5743551 (*TLR1*)	G (0.232)	0.396	0.140
	rs3804100 (*TLR2*)	C (0.081)	1.000	0.352
	rs4986790 (*TLR4*)	G (0.091)	0.572	0.077
	rs3764880 (*TLR8*)	G (0.213)	0.329 **	0.150
EUR	rs5743551 (*TLR1*)	G (0.285)	**0.016**	-
	rs3804100 (*TLR2*)	C (0.064)	1.000	
	rs4986790 (*TLR4*)	G (0.057)	0.207	
	rs3764880 (*TLR8*)	G (0.269)	0.566 **	

* Haldane Exact test for the Hardy–Weinberg equilibrium (autosomal) using the SELOME *p*-value. ** Graffelman–Weir exact test for the Hardy–Weinberg equilibrium on the X-chromosome using the SELOME *p*-value.

**Table 5 diagnostics-13-00121-t005:** Significant differences between the Case 1 group and the control group.

SNP	Control	Allele 1 Freq.	Case 1	Allele 2 Freq.	OR CI95%	*p*
**rs5743551 (*TLR1*)**						
**Codominant**						
A/A	60	60.6	40	52.6	1.00	0.045
A/G	32	32.3	35	46.1	1.64 (0.88–3.06)	
G/G	7	7.1	1	1.3	0.21 (0.03–1.81)	
**Recessive**						
A/A-A/G	92	92.9	75	98.7	1.00	0.052
G/G	7	7.1	1	1.3	0.18 (0.02–1.46)	
**rs3764880 (*TLR8*)**						
**Recessive**						
A/A-A/G	94	94.9	57	75.0	1.00	0.00012
G/G	5	5.1	19	25.0	6.27 (2.22–17.71)	

## Data Availability

The data presented in this study are available from the corresponding author upon request. The data are not publicly available in accordance with our ethical policy.

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
