# Peer review of "TLRs Gene Polymorphisms Associated with Pneumonia before and during COVID-19 Pandemic"

_diagnostics, 2022, doi:10.3390/diagnostics13010121_

Round 1
Reviewer 1 Report
This manuscript showed evidence of TLR polymorphism among pneumonia, COVID-19, and healthy groups. However, some comments to improve their manuscript appear below:
1. Line 46: enterobacteria is Gram-negative bacteria and the author is already mentioned, so, no need to refer. I recommend to delete it. Another, the author forget to include fungi which is an organism cause pneumonia.
2. Flow chart of study design is required for easy understanding of the reader.
3. Line 84: In case of COVID group, were all participant pneumonia?
4. Line 106-108: SNP selection is too short, please provide more detail how to select? what is criteria?
5. The author should summarize "take-home message/finding" at the end of each result subtopic. This will make the reader easily understand and get idea.
6. Line 174: control group? after Case 1, Case 2.
7. What is discussion of TLR2 that the author showed in abstract that " COVID19 group.....significant results are obtained for rs3804100 (TLR2)".
8. Conclusion is required.
Author Response
Dear reviewer,
Thank you for the detailed review of our article, then we have tried to give detailed answers to all the comments
Point 1: Line 46: enterobacteria is Gram-negative bacteria and the author is already mentioned, so, no need to refer. I recommend to delete it. Another, the author forget to include fungi which is an organism cause pneumonia.
Response 1: Thank you, we've prooved this part.
Point 2: Flow chart of study design is required for easy understanding of the reader.
Response 2: We've add flow chart of study design in materials and method section.
Point 3: Line 84: In case of COVID group, were all participant pneumonia?
Response 3: Yes, pneumonia were diagnosted for all participants in Case 2 (COVID) group. We include the following sentence in the text: ”Case 2 samples were collected from patients with bacterial pneumonia developed on the background of COVID-19.”
Point 4: Line 106-108: SNP selection is too short, please provide more detail how to select? what is criteria?
Response 4: In this study, we chose SNPs from studies that have been replicated. Since SNP's were selected on the basis of already published data on their association with the course of infectious processes, including respiratory diseases, additional information on SNP's selection has been added to the "Introduction" section.
Point 5: The author should summarize "take-home message/finding" at the end of each result subtopic. This will make the reader easily understand and get idea.
Response 5: Thank you, we’ve summarized our findings below your recommendation.
Point 6: Line 174: control group? after Case 1, Case 2.
Response 6: Thank you, we've add it in the text.
Point 7: What is discussion of TLR2 that the author showed in abstract that " COVID19 group.....significant results are obtained for rs3804100 (TLR2)".
Response 7: Thank you for notice, we clarified this part and delete these sentence.
Point 8: Conclusion is required.
Response 8: We include the conclusion section and submitted proofreading corrections.
Sertificare of proofreading corrections â„–56571

Reviewer 2 Report
The manuscript contains serious flaws, the results are poorly presented, hard to understand and the writing style requires major improvement. Discussion section contains little referral to previously published work, whereas the introduction does not provide sufficient insight into the rationale of the study. The following comments should be considered when re-submitting a new version of the manuscript:
1. Lines 49-51: please add references for the Russian guidelines.
2. There is no obvious link between SNPs of TLRs and pneumonia in the introduction section. Furthermore, justification of the choice of SNPs is not provided (also related to line 107, were references are lacking- I suggest moving these information though in the introduction section and removing the table format).
3. The study designchapter does not provide any information regarding the enrollment period.
4. If bacteria was isolated from tracheobronchial aspirate, should the reader understand that all the patients enrolled in the study were hospitalized in intensive care units? Please detail how tracheobronchial aspirate collection was conducted.
5. What did the microbial examination of the blood imply (line 102)?
6. Line 103: please add references for the PORT and SOFA scales.
7. Mean age should be expressed in a value with at least 2 decimals +/- SDs, even among the text.
8. Line 143: what does the EUR population represent?
9. Considering the age and gender discrepancy between study group and controls, I think the choice of the control group was unfortunate.
10. MAF acronym is not mentioned in the legend of table 4.
11. Table 4 contains no data regarding the EUR population (see the title).
12. Why were only part of the SNPs found in supplementary tables 1 and 2 analyzed in table 4?
13. Supplementary tables have no title.
14. The results are overall poorly represented, not detailed enough among the text and are hard to comprehend for a reader that does not specialize in genetics.
15. The manuscript lacks a clear, separate conclusion.
16. Discussion section is poor, with little references made to previous findings regarding the role of the analyzed TLRs in respiratory tract infections/pneumonia.
17. The writing style should be improved, not only within the manuscript, but also within the abstract which contains very many sentences which are not linked one to the other.
18. English language requires corrections.
Author Response
Dear reviewer,
Thank you for the detailed review of our article, then we have tried to give detailed answers to all the comments
Point 1: Lines 49-51: please add references for the Russian guidelines.
Response 1: Thank you for your notice, we’ve add references.
Point 2: There is no obvious link between SNPs of TLRs and pneumonia in the introduction section. Furthermore, justification of the choice of SNPs is not provided (also related to line 107, were references are lacking- I suggest moving these information though in the introduction section and removing the table format).
Response 2: Thank you, moved this part to introduction section, but for more obvious visualization we left the table in the materials and methods, changing the maintenance slightly.
Point 3: The study designchapter does not provide any information regarding the enrollment period.
Response 3: We've add the flow chart on this part of materials and methods section and provided this information.
Point 4: If bacteria was isolated from tracheobronchial aspirate, should the reader understand that all the patients enrolled in the study were hospitalized in intensive care units? Please detail how tracheobronchial aspirate collection was conducted.
Response 4: Thank you, yes, all patients were hospitalized in intensive care units, we’ve include this information in text. During the collection of material, methodological recommendations approved in Russian Federation were used. Unfortunately, the document is only in Russian. If necessary, we can translate the part of this document into English on request. We’ve added the information about tracheobronchial aspirate collection in materials and methods section.
Point 5: What did the microbial examination of the blood imply (line 102)?
Response 5: Thank you for your notice, we’ve include this information in text.
Point 6: Line 103: please add references for the PORT and SOFA scales.
Response 6: Thank you for your notice, we’ve add references.
Point 7: Mean age should be expressed in a value with at least 2 decimals +/- SDs, even among the text.
Response 7: Thanks for the comment, we’ve corrected it.
Point 8: Line 143: what does the EUR population represent?
Response 8: The EUR population characteristics derived from the 1,000 genomes project, which include genotype data from five caucasoid population - CEU (Utah Residents (CEPH) with Northern and Western European ancestry), FIN (Finnish in Finland), GBR (British in England and Scotland), IBS (Iberian population in Spain) and TSI (Toscani in Italia). We’ve include this information in materials and methods section.
Point 9: Considering the age and gender discrepancy between study group and controls, I think the choice of the control group was unfortunate.
Response 9: We fully agree with this comment. Due to the fact that our study had a pilot design and didn't include enougth samples for age/gender adjustment, we haven't conducted additional analyses. We described this limitation of our study in the conclusion section. We will certainly take this into account for future work.
Point 10: MAF acronym is not mentioned in the legend of table 4.
Response 10: Thank you for notice, we’ve include acronym in the legend of table 4.
Point 11: Table 4 contains no data regarding the EUR population (see the title).
Response 11: Thank you for notice, we’ve include data for EUR population in the table.
Point 12: Why were only part of the SNPs found in supplementary tables 1 and 2 analyzed in table 4?
Response 12: We've provide only significant results in the main text of our article, Supplementary files include information from all SNPs.
Point 13: Supplementary tables have no title.
Response 13: Thank you for notice, we've add titles for Supplementary tables.
Point 14: The results are overall poorly represented, not detailed enough among the text and are hard to comprehend for a reader that does not specialize in genetics.
Response 14: We have added a more detailed description of the results obtained and tried to make them more understandable.
Point 15: The manuscript lacks a clear, separate conclusion.
Response 15: We've added a conclusion section and include take-home finding at the end of each result subtopic.
Point 16: Discussion section is poor, with little references made to previous findings regarding the role of the analyzed TLRs in respiratory tract infections/pneumonia.
Response 16: We've expand more references to previous findings.
Point 17: The writing style should be improved, not only within the manuscript, but also within the abstract which contains very many sentences which are not linked one to the other.
Response 17: Thank you, we've improved the writing style.
Point 18: English language requires corrections.
Response 18: Thank you for your comment, we submitted the manuscript for proofreading.

Round 2
Reviewer 2 Report
1. Please rewrite the first paragraph of the conclusion, as it is very hard to understand.
2. I still believe that discussions should be expanded and improved.
3. English language and style still require corrections.
Author Response
Thank you for your detailed consideration of our manuscript. We have tried to make the manuscript clearer and easier to read.
Point 1: Please rewrite the first paragraph of the conclusion, as it is very hard to understand.
Response 1: We’ve rewrite the conclusion to make it more understandable.
Point 2: I still believe that discussions should be expanded and improved.
Response 2: We’ve tried to improve and expand more detailed information in discussion section. We tried to make it more comprehensive without deviating from the topic.
Point 3: English language and style still require corrections.
Response 3: Thank you for your comment, we submitted the manuscript for proofreading. You can find the proofreading sertificate for this article in attchment. If it necessary, we are ready to resend the manucript to the proofreading by native english editor.

Round 3
Reviewer 2 Report
The authors have successfully adhered to my recommendations.